# Cryo-EM reveals species-specific components within the *Helicobacter pylori* Cag type IV secretion system core complex

Michael J Sheedlo[1†], Jeong Min Chung[2†], Neha Sawhney[3], Clarissa L Durie[2], Timothy L Cover[1,3,4*], Melanie D Ohi[2,5*], D Borden Lacy[1,4*]

[1]Department of Pathology, Microbiology, and Immunology, Vanderbilt University Medical Center, Nashville, United States; [2]Life Sciences Institute, University of Michigan, Ann Arbor, United States; [3]Department of Medicine, Vanderbilt University School of Medicine, Nashville, United States; [4]Veterans Affairs Tennessee Valley Healthcare System, Nashville, United States; [5]Department of Cell and Developmental Biology, University of Michigan, Ann Arbor, United States

*For correspondence:
timothy.l.cover@vumc.org (TLC);
mohi@umich.edu (MDO);
borden.lacy@vanderbilt.edu (DBL)

[†]These authors contributed equally to this work

Competing interests: The authors declare that no competing interests exist.

**Abstract** The pathogenesis of *Helicobacter pylori*-associated gastric cancer is dependent on delivery of CagA into host cells through a type IV secretion system (T4SS). The *H. pylori* Cag T4SS includes a large membrane-spanning core complex containing five proteins, organized into an outer membrane cap (OMC), a periplasmic ring (PR) and a stalk. Here, we report cryo-EM reconstructions of a core complex lacking Cag3 and an improved map of the wild-type complex. We define the structures of two unique species-specific components (Cag3 and CagM) and show that Cag3 is structurally similar to CagT. Unexpectedly, components of the OMC are organized in a 1:1:2:2:5 molar ratio (CagY:CagX:CagT:CagM:Cag3). CagX and CagY are components of both the OMC and the PR and bridge the symmetry mismatch between these regions. These results reveal that assembly of the *H. pylori* T4SS core complex is dependent on incorporation of interwoven species-specific components.

## Introduction

*H. pylori* colonizes the stomach in about half of the world's human population, and its presence in the stomach is the strongest known risk factor for gastric cancer (*Uemura et al., 2001*). *H. pylori* strains containing a pathogenicity island (*cag* PAI), encoding CagA (a secreted effector protein) and the Cag T4SS, are associated with a significantly higher incidence of gastric cancer than that associated with strains lacking the *cag* PAI (*Blaser et al., 1995*; *Cover, 2016*). Gastric cancer is the third leading cause of cancer-related death worldwide, and there are nearly one million new gastric carcinoma cases annually (*Plummer et al., 2016*). Therefore, understanding the molecular organization of the Cag T4SS is an important goal.

Bacterial T4SSs are versatile molecular machines that can mediate a wide variety of functions, including horizontal transfer of DNA among bacteria (conjugation) and delivery of effector proteins into eukaryotic cells (*Galán and Waksman, 2018*; *Grohmann et al., 2018*; *Waksman, 2019*). T4SSs in Gram-negative bacteria are comprised of an inner membrane complex, a membrane-spanning core complex that extends from the inner membrane to the outer membrane, and in some systems, an extracellular pilus (*Galán and Waksman, 2018*; *Grohmann et al., 2018*; *Waksman, 2019*). T4SSs vary in complexity, ranging from 'minimized' or 'prototype' systems that contain about 12 components (VirB1-11 and VirD4), as found in *Agrobacterium tumefaciens*, *Xanthomonas citri*, or

conjugation systems, to expanded systems containing more than 20 components (*Bhatty et al., 2013*). Expanded T4SSs contain unique species-specific components and are much larger in size than their minimized counterparts. Examples of the larger, 'expanded' T4SSs include the Dot/Icm T4SSs of *Legionella pneumophila* and *Coxiella* and the Cag T4SS of *H. pylori,* each of which transports effector proteins into host cells (*Chetrit et al., 2018*; *Chung et al., 2019*; *Frick-Cheng et al., 2016*; *Ghosal et al., 2019*). The *Legionella* Dot/Icm translocates more than 300 effector proteins, thereby allowing intracellular bacterial replication (*Qiu and Luo, 2017*). In contrast, the *H. pylori* Cag T4SS is only known to deliver one effector protein, CagA (a bacterial oncoprotein), into gastric cells, resulting in altered cell signaling (*Cover et al., 2020*).

Structural studies of minimized T4SSs have revealed a core complex (typically ~250 Å in width and ~150 Å in height) that contains three proteins (VirB7, VirB9, and VirB10) (*Chandran et al., 2009*; *Fronzes et al., 2009*; *Low et al., 2014*; *Sgro et al., 2018*). The *H. pylori* Cag T4SS core complex is a much larger mushroom-shaped assembly (~400 Å in width and ~250 Å in height) that contains structural homologs of the proteins found in minimized systems [CagT (VirB7), CagX (VirB9), and CagY (VirB10)] along with additional proteins (Cag3 and CagM) that lack homologs in other bacterial species (*Chung et al., 2019*; *Frick-Cheng et al., 2016*). The Cag T4SS core complex has been described as consisting of three distinct structural features: the outer membrane cap (OMC) consisting of an outer-layer (O-layer) and inner-layer (I-layer), a periplasmic ring (PR), and a stalk (*Chang et al., 2018*; *Chung et al., 2019*; *Hu et al., 2019*). There is an apparent symmetry mismatch between the 14-fold-symmetric OMC and the 17-fold symmetric PR (*Figure 1a*; *Chung et al., 2019*). Our recent sub-4Å structure of the Cag T4SS revealed that central portions of the OMC are composed of CagT, CagX, and CagY (VirB7, VirB9, and VirB10 homologs), but the resolution of the map

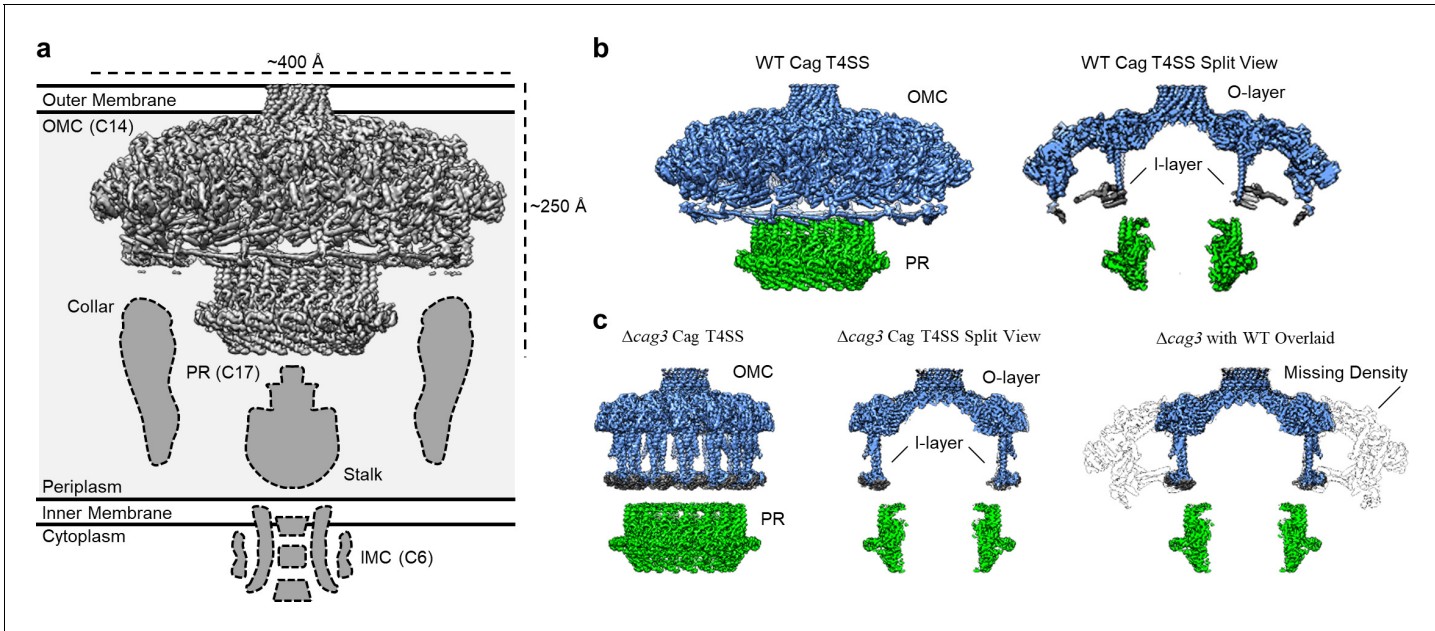

**Figure 1.** Comparison between maps of the wild-type (WT) and ΔCag3 Cag T4SS core complex. (**a**) The Cag T4SS spans the inner membrane and outer membrane and consists of distinct features with differing symmetry: the OMC (C14 symmetry), the PR (C17 symmetry), an inner-membrane complex (IMC, C6 symmetry), and the stalk and collar with unknown symmetries. (**b**) In the reconstruction of the WT Cag T4SS, we observe the O-layer of the OMC (shown in blue), the I-layer of the OMC (shown in gray), and the PR (shown in green). Left panel, WT Cag T4SS density map; Right panel, central slice of the WT density map. (**c**) All of these features are observed in the ΔCag3 Cag T4SS map (shown in the same colors as panel b), but peripheral regions of the OMC are missing in the ΔCag3 Cag T4SS map (shown in white). Left panel, ΔCag3 Cag T4SS density map; middle panel, central slice of ΔCag3 Cag T4SS density map; right panel, overlaid central slices of WT (grey) and ΔCag3 (blue and green) Cag T4SS density maps.

The online version of this article includes the following figure supplement(s) for figure 1:

**Figure supplement 1.** Flow chart of cryo-EM processing steps.

**Figure supplement 2.** Cryo-EM processing of the ΔCag3 Cag T4SS core complex.

**Figure supplement 3.** Cryo-EM processing of the WT Cag T4SS core complex.

**Figure supplement 4.** Analysis of Cag T4SS core complex preparations.

did not allow us to build molecular models for the entire complex (*Chung et al., 2019*). Specifically, we could not determine the molecular composition of the PR or define the locations of Cag3 or CagM. In the current study, we undertook a further analysis of single particle cryo-EM data for the wild-type Cag T4SS core complex as well as analysis of a mutant form of the core complex. These efforts allow us to define the locations of Cag3 and CagM within the core complex, determine the molecular composition of the PR, and determine the stoichiometry of components within the OMC.

## Results

To build a more complete model of Cag T4SS organization and determine how the OMC and PR interact, we expanded our cryo-EM studies to analyze the core complex from a Cag3-deficient *H. pylori* strain (Δ*cag3*) (*Frick-Cheng et al., 2016*). We also implemented new data analysis techniques for processing previously collected cryo-EM data for wild-type (WT) core complexes (including per-particle defocus refinement and beam tilt estimation), which resulted in higher resolution maps (*Figure 1—figure supplements 1*, *2* and *3*, *Appendix 1—table 1*). When comparing the maps reconstructed from the WT and Δ*cag3* strains, peripheral components of the O-layer of the OMC are missing in the Δ*cag3* map. At least a portion of this peripheral density is likely composed of Cag3, since it is the component most notably absent in core complex preparations from the Δ*cag3* mutant [based on SDS-PAGE and mass spectrometry analyses (*Figure 1—figure supplement 4*)]. Notably, the PR remained intact in the Δ*cag3* map, and weak density was still observed within the I-layer (*Figure 1b,c*).

As described previously, CagT and the C-terminal portions of CagX (residues 349–514) and CagY (residues 1677–1909) are localized to central regions of the O-layer of the OMC (*Chung et al., 2019*). The increased resolution of the new maps allowed us to define the OMC asymmetric unit and determine the molecular composition of regions that were previously undefined (*Figure 1—figure supplement 2d*, *Figure 2a*, *Figure 2—figure supplements 1–6*, *Video 1*). Positioned adjacent and peripheral to the previously described CagT (CagT-1), we identified another copy of CagT which we call CagT-2 (*Figures 2a,b* and *3a*). We found that the two copies of CagT differ significantly in the conformations of both their N- and C-termini (*Figure 4*). In CagT-1, the N-terminus is an extended loop that is nestled against a portion of CagT-1 and CagX from the neighboring asymmetric unit, resulting in an interface similar to what was described for *X. citri* VirB7 and VirB9 (*Sgro et al., 2018*). Within CagT-2, the N-terminus adopts a different conformation, one in which it is wrapped back around the central, globular VirB7-like fold, forming a β-strand that completes a β-sheet within an adjacent protein that we designate as Cag3-1 (*Figure 4c–f*, *Video 2*). This rearrangement likely occurs due to changes in a stretch of amino acids (I44-I50) within the center of this loop (*Figure 4—figure supplement 1a–c*). While it is currently not clear what drives these conformational differences, it is possible that this rearrangement results from the different repertoire of binding partners mediated by this loop (*Figure 4—figure supplement 1d–f*). Despite the structural differences between CagT-1 and CagT-2, a putative lipidation site (C21) in both proteins (*McClain et al., 2020*) is placed in close proximity to the outer membrane (*Figure 4g*). The C-termini of both CagT molecules adopt extended conformations that differ in their overall direction and organization. In CagT-1, the C-terminal α-helices extend outward from the center of the map, forming contacts with CagT-1 Cag3-1, Cag3-2, and Cag3-4 within the OMC (*Figure 4h*; *Chung et al., 2019*). Conversely, the C-terminal α-helices of CagT-2 are connected by an apparently dynamic linker and are arranged such that they contact a distinct repertoire of OMC proteins, including Cag3-1, Cag3-2, and CagM-1 (*Figure 4i*). In both cases, the arrangement of these helical extensions deviates significantly from corresponding regions in the *X. citri* VirB7 structure, where the domain terminates shortly after the core VirB7 fold (*Sgro et al., 2018*; *Figure 4—figure supplement 2a*). Because of this, we suggest that the addition of these C-terminal α-helices is critical to maintaining the expanded architecture of the Cag T4SS OMC, as it has been previously shown that Cag3 cannot be effectively incorporated into core complexes lacking CagT (*Frick-Cheng et al., 2016*).

The orientation of the CagT-1 C-terminal α-helices differs significantly in the Δ*cag3* OMC reconstruction compared to their orientation in the wild-type OMC reconstruction. In contrast to the extended conformation of the CagT-1 α-helices in the WT OMC, in the Δ*cag3* OMC these helices are folded inward and pack against α-helices of CagX and CagY within the central chamber (*Figures 2c,d* and *5a*, *Figure 4—figure supplement 2b*). Notably, the second α-helix of the

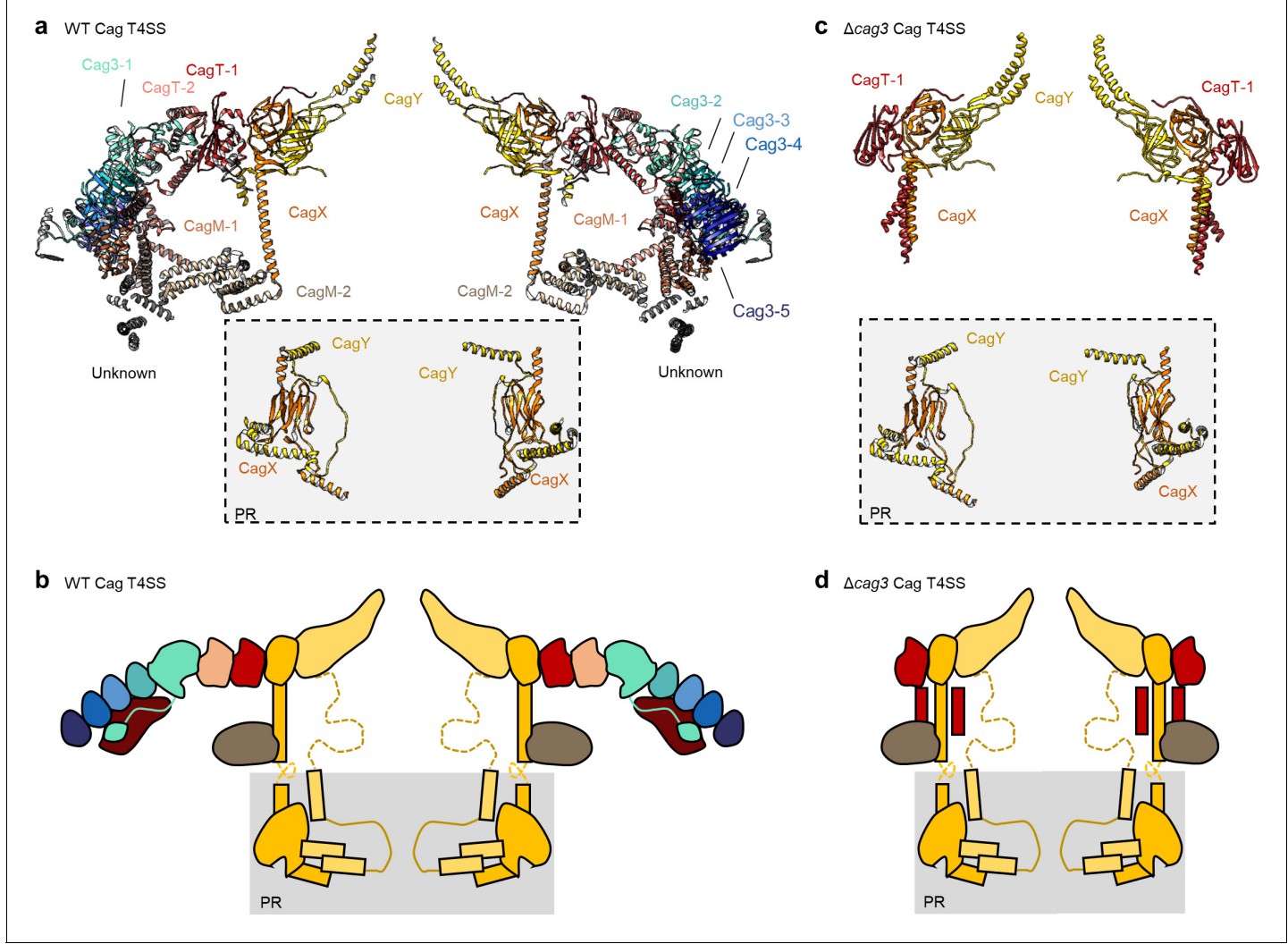

**Figure 2.** The asymmetric unit of the WT and ΔCag3 Cag T4SS core complex. (**a**) A cross-section model of the mapped portions of CagY (yellow), CagX (orange), CagT (red and salmon), CagM (brown and tan), and Cag3 (various shades of green and blue) in the OMC from the WT Cag T4SS. Portions of CagX and CagY are present in both the OMC and the PR (denoted by the gray box). (**b**) A cartoon representation of the WT Cag T4SS core complex proteins, colored as in panel a. The dotted lines represent densities that were not clearly seen in the density maps. (**c**) A cross-section model of the mapped portions of CagY (yellow), CagX (orange), and CagT (red) from the ΔCag3 Cag T4SS core complex. (**d**) A cartoon representation of the ΔCag3 Cag T4SS core complex proteins with CagY (yellow), CagX (orange), CagT-1 (red). The red rectangles represent the alternate conformation of the C-terminal helices of CagT-1. CagT-2 and Cag-3 were not observed within this map. The dashed lines represent densities that were not clearly seen in the density map.

The online version of this article includes the following figure supplement(s) for figure 2:

**Figure supplement 1.** Correlation of each component of the WT Cag T4SS to experimental maps.

**Figure supplement 2.** Correlation of each component of the ΔCag3 Cag T4SS to experimental maps.

**Figure supplement 3.** Correlation between the WT OMC cryo-EM map and models.

**Figure supplement 4.** Model-map correlation for each protein within the OMC.

**Figure supplement 5.** Quality of models within the OMC of the ΔCag3 T4SS.

**Figure supplement 6.** Quality of models within the PR of the WT and ΔCag3 maps.

C-terminal extension is not observed within the Δ*cag3* OMC map and may be flexible. We suspect that the C-terminal α-helices of CagT-1 adopt multiple conformations in the Δ*cag3* OMC, since additional components that would otherwise lock them into a single conformation are not present. In addition to the loss of peripheral density corresponding to Cag3 within the Δ*cag3* map, we observe no density for CagT-2 in the Δ*cag3* map. One possible explanation for the loss of CagT-2 is that it

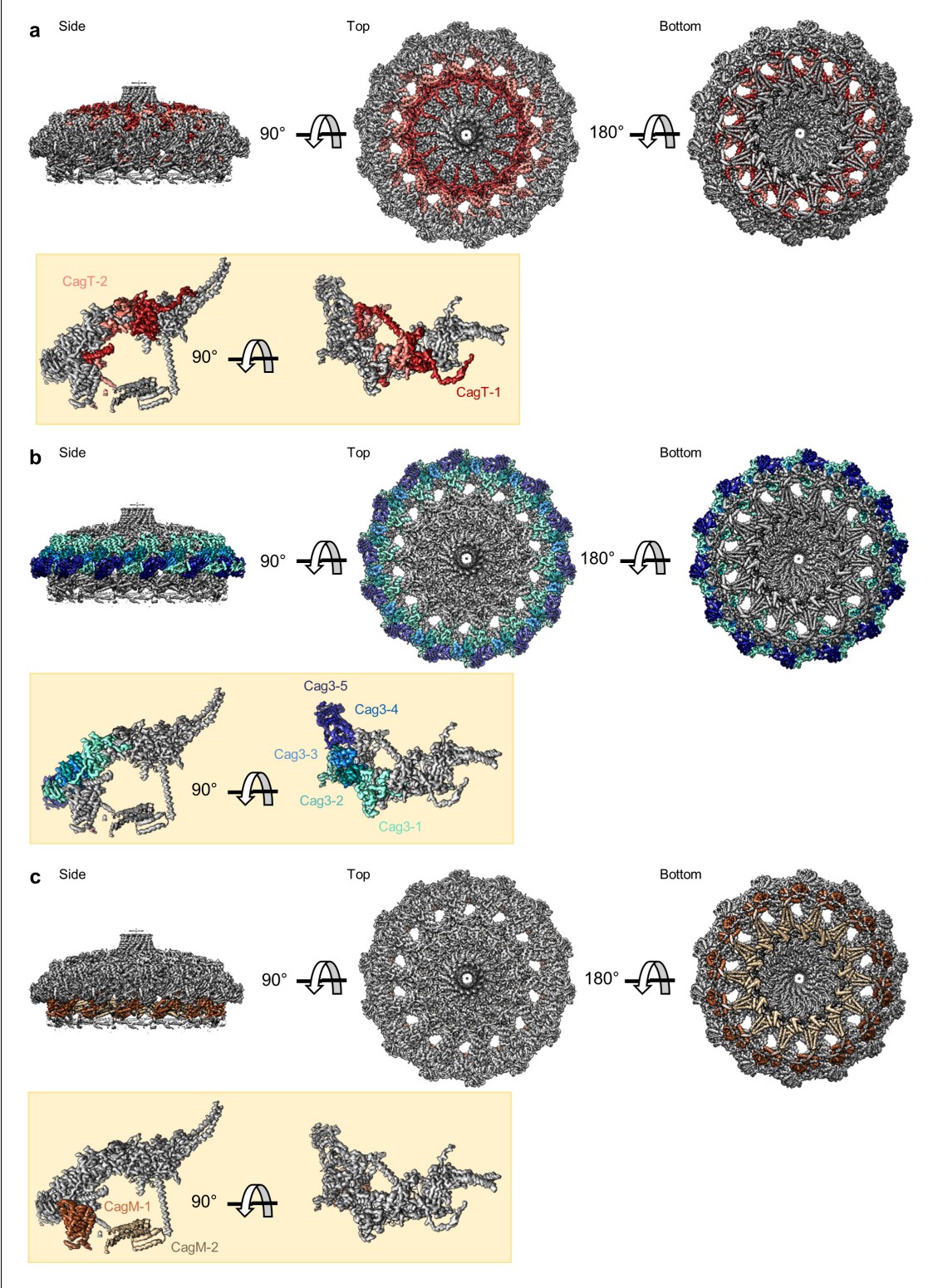

**Figure 3.** Locations of components within the OMC. (a) The O-layer of the OMC contains two copies of CagT (CagT-1 and CagT-2) (shown in red and salmon). The two copies are localized within the center of the asymmetric unit (inset) as shown in red and salmon. (b) Cag3 comprises a significant portion of the O-layer of the Cag T4SS, as shown in blue and green. Within the asymmetric unit (inset), we observe five copies of Cag3 (denoted Cag3-1 through Cag3-5, shown in various shades of blue and green). (c) Within the I-layer of the OMC, there are two similar folds, each defined as CagM. Within the asymmetric unit, we observe two copies (inset) that are colored in brown and tan.

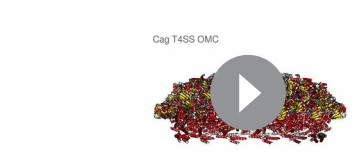

**Video 1.** The Cag T4SS OMC.
https://elifesciences.org/articles/59495#video1

cannot effectively fold when Cag3 is absent, leading to dissociation of the complex or unfolding of the protein. These results are largely in agreement with previous cryo-electron tomography (Cryo-ET) studies, which noted the loss of most of the O-layer of the OMC in the Δ*cag3* strain (*Hu et al., 2019*). CagX and CagY adopt nearly identical conformations in the ΔCag3 mutant core complex when compared to the WT structure (RMSDs of 0.5 Å and 0.3 Å, respectively) (*Figure 5b,c*). This is consistent with a proposal that assembly of the Cag T4SS is initiated by positioning CagX and CagY in association with the outer membrane (*Hu et al., 2019*).

Further analysis revealed five copies of Cag3 within each asymmetric unit (named Cag3-1 through Cag3-5, based on their proximity to the central channel) (*Figure 2a,b Figure 3b*). All five copies of Cag3 are found within the periphery of the OMC and adopt similar core folds (with RMSDs of 0.5–1.3 Å) that are connected via a highly interwoven architecture (*Figure 6a–c*). Although the resolution is lower towards the periphery of the map than near the center, we note that several Cag3 structural features are recognizable in all five copies of Cag3 within the cryo-EM density map (*Figure 6d–f*, *Figure 6—figure supplement 1*). The first copy of Cag3 (Cag3-1) contains the longest uninterrupted span of density (corresponding to residues 62–308) and consists of two globular domains (proximal and distal) connected by a long loop, without an intramolecular interface (*Figure 6a,b*). The proximal domain of Cag3-1 is positioned adjacent to CagT-2 within the center of the map and contains a loop (residues 181–204) that interacts with adjacent asymmetric units (*Figure 6g*). We predict that this loop may act as a lynchpin during assembly, since it forms the most extensive network of contacts within the center of the complex. Notably, the proximal domain of Cag3-1 consists of a 'core' fold that is similar to a corresponding fold in CagT and the related VirB7 homolog from *X. citri* (*Figure 6h*; *Sgro et al., 2018*). The distal domain of Cag3-1 is linked to the proximal domain through a linker that runs along a second copy of Cag3 (Cag3-2) and resides within the periphery of the map,~50 Å away from the proximal domain, tucked up next to another copy of Cag3 (Cag3-3) (*Figure 2a,b*). A distal domain was not seen in any of the other copies of Cag3, possibly due to the lower resolution at the periphery of the map or to inherent heterogeneity associated with this domain.

While the local resolution of the I-layer is lower than that of the O-layer, we observe several key features of the I-layer that are consistent with the sequence of CagM (*Figures 3c* and *7a*). We modeled a portion of CagM (residues 187–366) into peripheral density within the I-layer and observe an α-helical fold with two sub-domains: a 3-helix bundle at the N-terminus connected by a short loop to a 5-helix fold at the C-terminus. We also observe a second helical fold containing 9 α-helices, arranged in a nearly identical fashion as in CagM-1 within the center of the I-layer, albeit at lower resolution (*Figure 7b*). We fit the CagM sequence into this density and note that the refined structure (designated CagM-2) again contains two subdomains connected via a flexible hinge (*Figure 7c*).

When aligning each subdomain of CagM-1 with the corresponding subdomain of CagM-2, the RMSD for each is ~1.1 Å, supporting our assignment of this portion of the map as a second copy of CagM (*Figure 7d,e*). The difference in the orientation of the subdomains in CagM-1 and CagM-2 is the result of unique contacts mediated by the C-terminal subdomains (*Figure 7f,g*).

Based on these analyses, we conclude that the OMC proteins exist at a stoichiometry of 1:1:2:2:5 (CagX:CagY:CagM:CagT:Cag3). This stoichiometry helps explain the large size of the Cag T4SS core complex, since core complexes in characterized minimized systems contain only

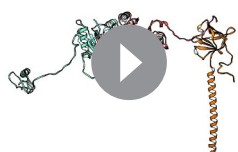

**Video 2.** Differences between CagT-1 and CagT-2 in the Cag T4SS asymmetric unit.
https://elifesciences.org/articles/59495#video2

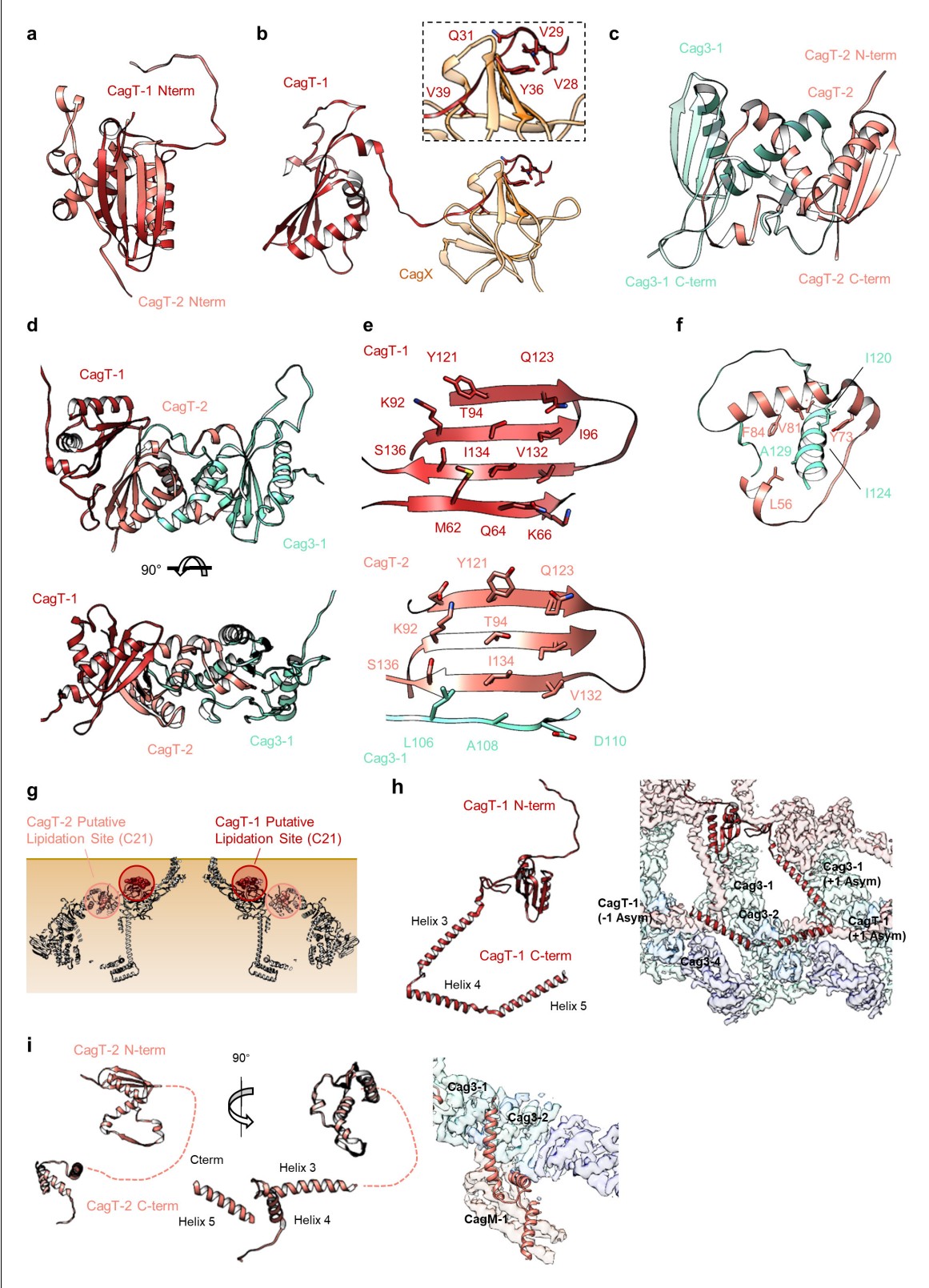

**Figure 4.** The N- and C-termini from CagT-1 and CagT-2 are positioned differently. (**a–c**) The N-terminus of CagT-1 extends inward toward the center of the map and interacts with CagX from the next asymmetric unit. The residues that contribute to this interaction are largely hydrophobic, as indicated in the inset panel. CagT-2 differs from CagT-1 in that the N-terminus of the protein extends outward toward the periphery of the map, forming the last strand of a β-sheet with Cag3-1. (**d**) The three proteins (CagT-1, CagT-2 and Cag3-1) have an interwoven architecture. (**e**) The interface that is formed

*Figure 4 continued on next page*

*Figure 4 continued*

between the three proteins consists of two β-sheets that include strands from all three molecules. (f) The interface of CagT-2 and Cag3-1 is a pair of α-helices that bury hydrophobic residues within the interface. (g) The position of the N-terminal loops of CagT-1 (red) and CagT-2 (salmon) are such that the putative lipidation sites are near the outer membrane. (h) The C-terminal α-helices of CagT-1 adopt an extended conformation (left) and interact with Cag3-1, Cag3-2, and Cag3-4 within the same asymmetric unit and Cag3-1 and CagT-1 in neighboring asymmetric units (right). (i) The C-terminal α-helices of CagT-2 are connected by an apparently flexible linker (left) and interact with Cag3-1, Cag3-2, and CagM-1 (right).

The online version of this article includes the following figure supplement(s) for figure 4:

**Figure supplement 1.** The N-terminal loop of CagT is observed in two different conformations.
**Figure supplement 2.** The C-terminal helices of CagT mediate interactions within the OMC.

single copies of each component within the asymmetric unit (*Fronzes et al., 2009*; *Sgro et al., 2018*). Illustrating this point, we note that the inclusion of 5 copies of Cag3 within the asymmetric unit results in 70 copies of Cag3 incorporated into the fully assembled Cag T4SS OMC. In total, the Cag T4SS OMC contains 154 different polypeptide chains. In comparison, prototype core complexes contain a total of 42 different polypeptide chains (14 copies each of VirB7, VirB9 and VirB10).

The resolution of the PR within the Δ*cag3* map is higher than what was previously determined for the wild-type complex (*Figure 1—figure supplement 2j,k*; *Chung et al., 2019*). Because of this advance, it is possible to identify the two components of the PR as portions of CagX (residues 32–130, 261–323) and CagY (residues 1469–1603). The PR is mostly comprised of a globular domain formed by an N-terminal portion of CagX, which starts from the cytoplasm-facing side of the PR, wraps back and forth to form two β-sheets, and ends in an α-helix pointed towards the OMC (*Figure 8a*, *Video 3*). The N-terminal domain of CagX is homologous to the N-terminal domain of *X. citri* VirB9 (RMSD of 0.8 Å, PDB 6GYB) (*Figure 8b–d*; *Sgro et al., 2018*). The C-terminal domain of CagX is also highly similar to its *X. citri* counterpart (RMSD of 0.6 Å, PDB 6GYB) (*Figure 8e,f*). The major difference between the two proteins is the presence of an insertion within CagX that forms a long α-helix in the center of the molecule (*Figure 8d,g*). Because of this feature, the N- and C-terminal domains of *H. pylori* CagX are positioned ~200 Å apart (within the PR and OMC, respectively), whereas the corresponding domains of *X. citri* VirB9 are ~80 Å apart (*Figure 8h*).

Wrapped around the PR portion of CagX is a segment of CagY that starts from the cytoplasm-facing side of the PR and forms four α-helices that are connected by loops that vary in length (ranging from 3 to 41 residues) (*Figure 8i*). This is similar to a described interaction between

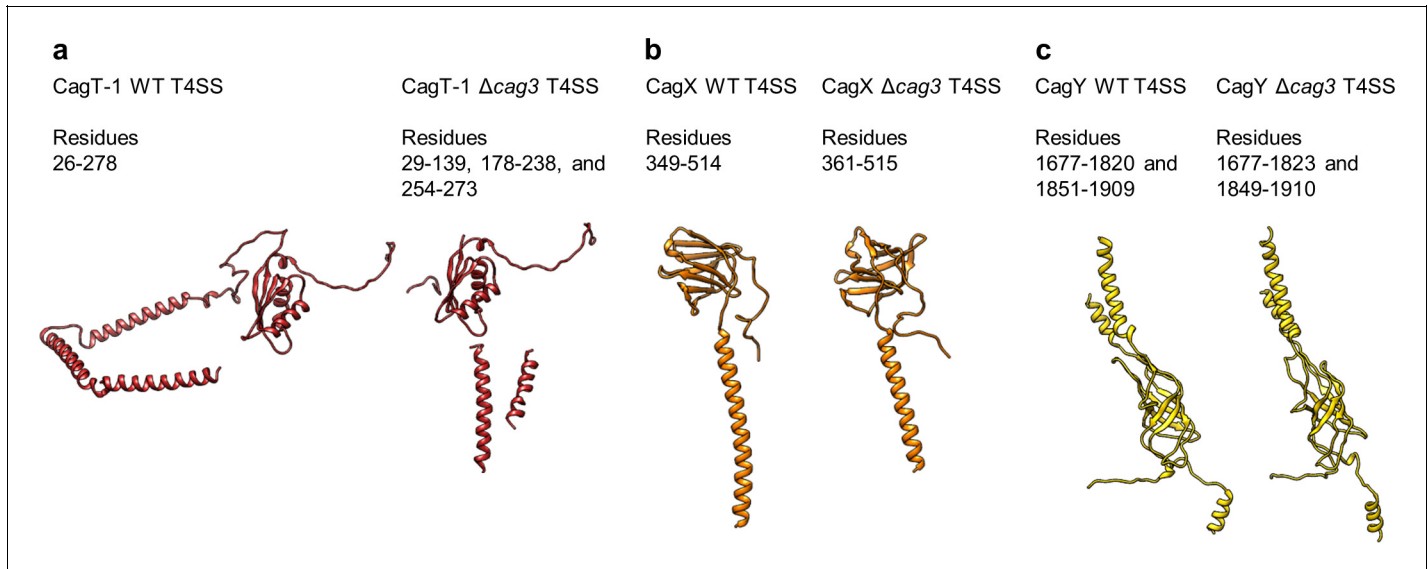

**a**
CagT-1 WT T4SS

Residues
26-278

CagT-1 Δ*cag3* T4SS

Residues
29-139, 178-238, and 254-273

**b**
CagX WT T4SS

Residues
349-514

CagX Δ*cag3* T4SS

Residues
361-515

**c**
CagY WT T4SS

Residues
1677-1820 and 1851-1909

CagY Δ*cag3* T4SS

Residues
1677-1823 and 1849-1910

**Figure 5.** Conservation of protein structures in WT and ΔCag3 Cag T4SS complexes. (a) Despite global differences between the overall WT and Δ*cag3* maps, the structure of CagT-1 shares a similar core VirB7-like fold in both maps. (b–c) Similarly, CagX and CagY adopt nearly identical orientations within both maps. The illustrated portions of CagX and CagY are the domains localized to the OMC.

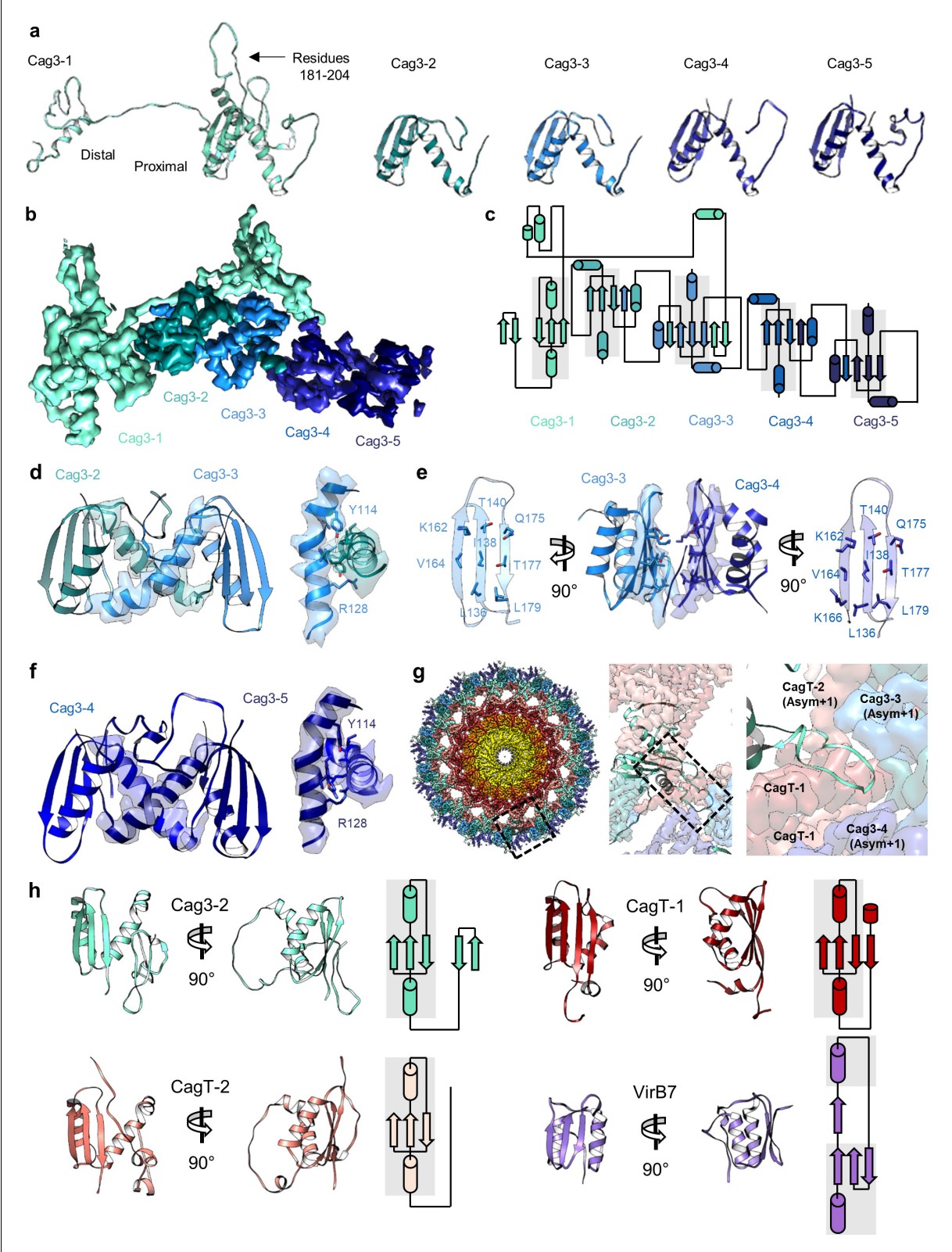

**Figure 6.** There are five copies of Cag3 within the OMC asymmetric unit. (a) Five different molecules of Cag3 are present within the asymmetric unit. The general architecture of Cag3 can be described as two domains, the proximal domain (residues 62–232) and the distal domain (residues 252–309). (b) All five of the Cag3 proteins share a heavily interwoven architecture in which β-sheets are formed between adjacent molecules within the asymmetric unit. (c) A topology diagram showing the general architecture of all copies of Cag3 within the asymmetric unit. (d) The interface of Cag3-2 (cyan) and

*Figure 6 continued on next page*

*Figure 6 continued*

Cag3-3 (light blue) is formed predominantly by two helices with contact mediated by Y114 and R128 of each molecule. (e) The interface of Cag3-3 (light blue) and Cag3-4 (blue) is dramatically different from the other Cag3 interfaces and includes an extensive hydrophobic interaction that is formed by two adjacent beta sheets. (f) Cag3-4 (blue) and Cag3-5 (navy blue) share a similar interface as Cag3-2 and Cag3-3, as shown above. (g) A view of the Cag T4SS is shown from the top-down (left) and indicates the position of a loop within the Cag3-1 proximal domain (amino acids 181–204, black box) that mediates contacts between asymmetric units. (h) The proximal domain of Cag3 contains a fold that is structurally similar to folds within CagT-1 (shown in red), CagT-2 (shown in salmon) and *X. citri* VirB7 (PDB 6GYB, shown in purple).

The online version of this article includes the following figure supplement(s) for figure 6:

**Figure supplement 1.** The repetitive β-sheet structure of CagT and Cag3.

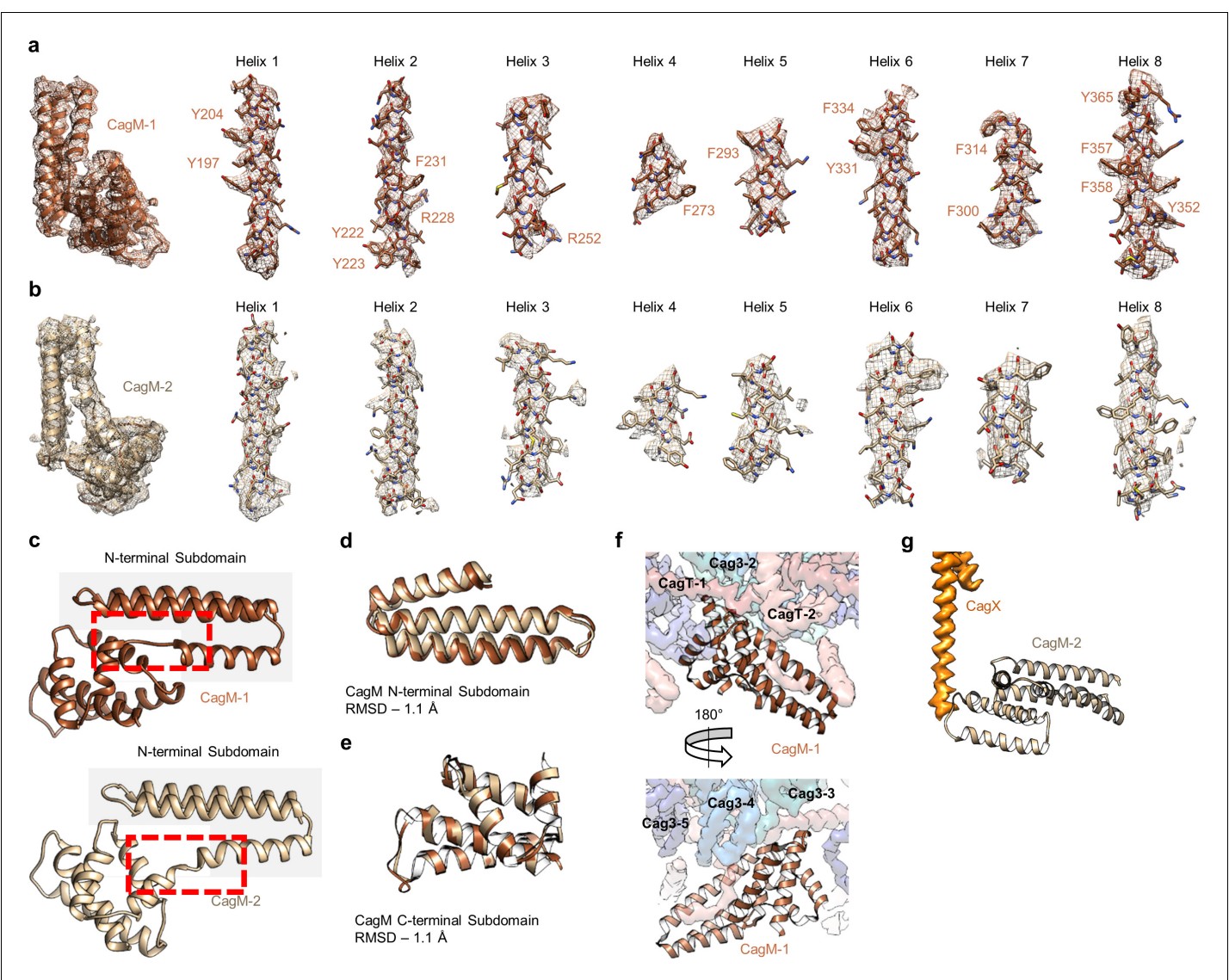

**Figure 7.** Localization of CagM within the I-layer of the OMC. (a) We modeled a single domain of CagM-1, consisting of 8 helices within the I-layer of the OMC. Representative density for all eight helices is shown with landmark residues indicated. (b) Although the local resolution within the I-layer is notably lower than that of the O-layer, several structural features within the I-layer density are consistent with the sequence of CagM (designated CagM-1). The correlation of all 8 helices of CagM-2 to the experimental map is shown. (c) The two subdomains within CagM are connected by a hinge (noted by the red dotted line). When the CagM-1 subdomains are aligned independently, we note RMSDs of 1.1 Å for both the N-terminal subdomain (d) and the C-terminal domain (e). The differences in sub-domain orientation are likely the result of differences in interacting partners of CagM-1 (f) and CagM-2 (g) within the OMC.

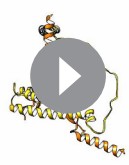

**Video 3.** Arrangement of CagX and CagY within the PR of the Cag T4SS.
https://elifesciences.org/articles/59495#video3

a corresponding part of VirB10 and the N-terminal domain of VirB9 in the *X. citri* T4SS (*Sgro et al., 2018*). In both cases, a segment of CagY/VirB10 in this region is linked to the C-terminal portion of the protein through a linker which is not directly observed in the data. Both CagY and *X. citri* VirB10 start with the N-terminus of this portion positioned near the periplasmic space and pass 'below' two protomers of CagX/VirB9. Both proteins then extend upward toward the outer membrane through the interior of the ring formed by the N-terminal domain of CagX/VirB9. The CagX-CagY interface in *H. pylori* is much larger than the corresponding VirB9-VirB10 interface, likely due to an expansion of CagY compared to VirB10 homologs in minimized systems (*Figure 8j*). The position of CagY within the PR could potentially allow the unresolved segment (residues 1604–1676) to span the inner chamber of the OMC to connect with a C-terminal domain of CagY in the OMC. We note that the portion of CagY visible in the PR is positioned so that the remaining N-terminal 1468 residues (not observed) likely extend into the periplasm, possibly contributing to the low resolution 'collar' and stalk densities (*Figure 8k*; *Hu et al., 2019*).

The identification of CagX and CagY within the PR was unexpected, as it suggests that each protein exists within both the OMC and PR, two distinct regions of the T4SS that differ in overall symmetry (*Figure 9a*). Although we cannot specifically trace either component across the symmetry mismatch, we note that the long helical expansion within CagX appears to bridge the OMC and PR. This model is supported by focused refinement of the connecting region between the OMC and PR in the WT map without imposition of symmetry, which shows 14 tubes of helical density connecting the OMC and PR (*Figure 9b*). Notably, only 14 copies of CagX within the PR appear to traverse the symmetry mismatch, leaving three domains within the PR without an obvious connection to the OMC (*Figure 9c*, *Video 4*). It is not clear if the density corresponding to these copies of CagX and CagY cannot be traced due to inherent flexibility within the respective C-terminal domains, if these additional protomers represent truncated versions of CagX and CagY, or if they represent uncharacterized structural homologs.

## Discussion

The Cag T4SS structure reported here represents the most comprehensive high-resolution analysis of the unique architectural features of an expanded T4SS to date. We have identified the positions of all five core complex proteins, including the previously uncharacterized Cag3 and CagM. A notable discovery is the structural conservation between central domains of CagT and Cag3, a result which could not have been predicted based on sequence comparisons alone. Our finding that the Cag T4SS incorporates multiple copies of Cag3, CagM and CagT into the OMC allows for a refined understanding of how this very large complex is assembled from only a few components. The current analysis also provides a better understanding of how the components of the Cag T4SS are arranged, highlighting important interactions among the newly described components. Specifically, interactions between CagT-1-CagT-2, CagT-1-Cag3-1, Cag3-Cag3, Cag3-CagM, and CagM-CagX have all been localized within this map. Although this study presents the first structural description of these interactions, it is interesting to note that many of these interactions have been previously detected by various biochemical methods. Specifically, interactions between Cag3-Cag3, Cag3-CagT, Cag3-CagM, CagX-CagM, and CagT-CagM were previous detected using a yeast two-hybrid screening

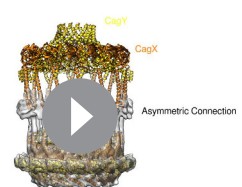

**Video 4.** Apparent Symmetry of CagX in the OMC and PR.
https://elifesciences.org/articles/59495#video4

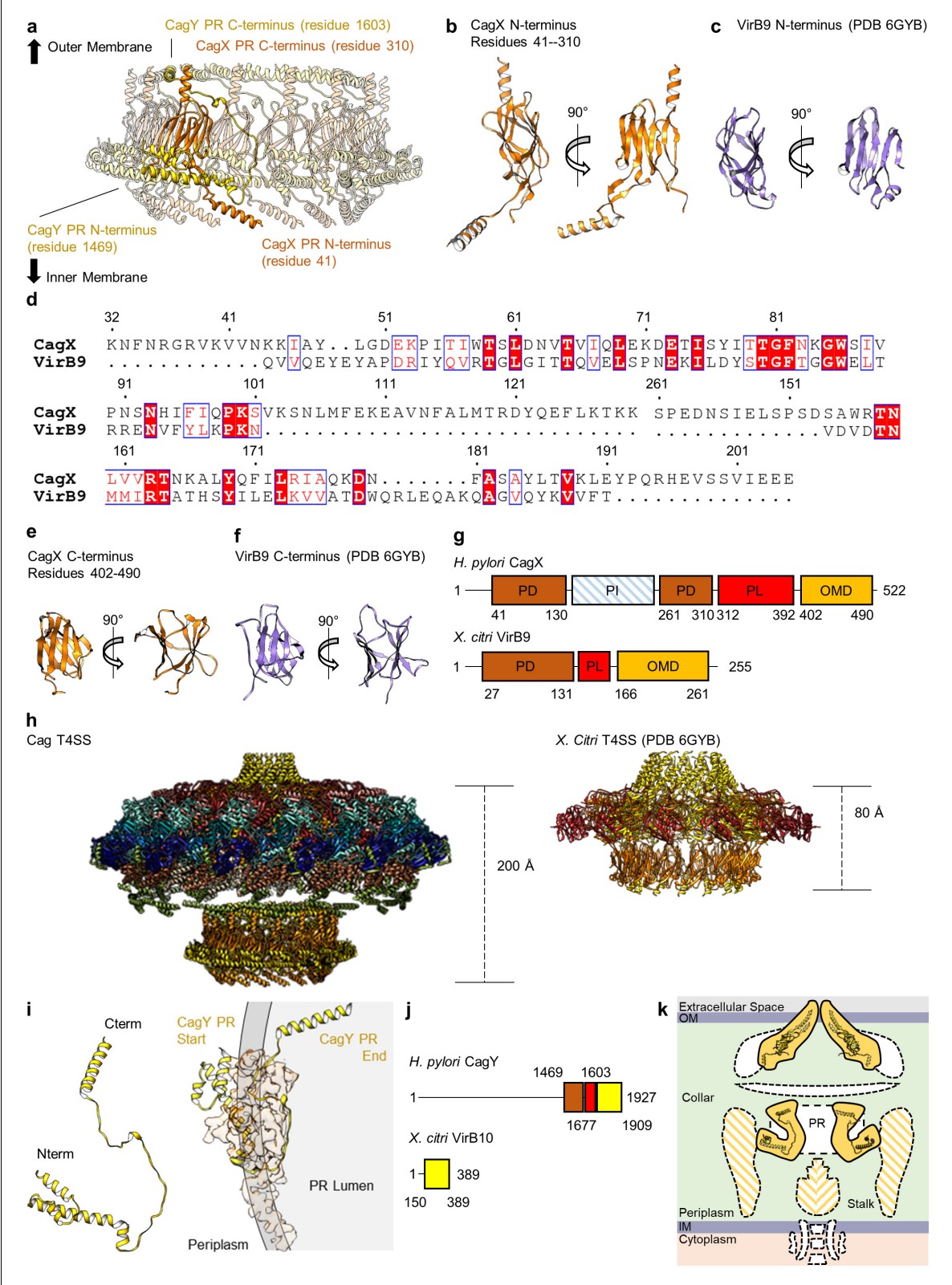

**Figure 8.** CagX and CagY comprise the PR. (**a**) The PR is comprised of N-terminal portions of CagX and a fragment of CagY. Both proteins start from the inner membrane side of the PR, form small globular folds, and extend upward toward the outer membrane. A portion of CagY within the PR wraps around CagX, starting from the periplasm and winding into the lumen of the PR. (**b**) The N-terminal domain of CagX (residues 41–310) is similar to that of VirB9 (**c**) from *X. citri* in both structure and sequence (**d**). The C-terminal domain of CagX (**e**) is similar to that of the C-terminal domain of VirB9 from

*Figure 8 continued on next page*

*Figure 8 continued*

*X. citri* (f). (g) The periplasmic and outer membrane domains (PD and OMD, respectively) are similar in structurally characterized VirB9 homologs, though they are separated by a periplasmic linker (PL) that is variable in length. CagX contains an additional insertion (residues 102–153, periplasmic insertion or PI) that is unique when compared to other homologs. The structure corresponding to the PI was not observed within any of our cryo-EM reconstructions. (h) The spacing of the two CagX/VirB9 domains varies depending on the organism and appears to be highly dependent on the length of the periplasmic linker (CagX and VirB9 are shown in orange). (i) The segment of CagY within the PR (residues 1469–1603) adopts a highly elongated fold that consists of four α-helices (left). The periplasmic portion of CagY (as shown on the left) starts within the periplasm and wraps around the globular domain of CagX (shown in orange), eventually ending in the lumen of the PR (gray, right). (j) The periplasmic segment of CagY, which we call the periplasmic domain (PD, brown), is unique to CagY and is not present in other VirB10 homologs such as VirB10 from *X. citri* (yellow indicates a VirB10 like domain, and red represents the unstructured linker). (k) The N-terminus of CagY was not observed within any of these cryo-EM reconstructions. The N-terminal portion of CagY in the model that was constructed (about residue 1469) is positioned so that it might continue outward into the periplasmic space, possibly contributing to the structural feature known as the collar, as well as the stalk.

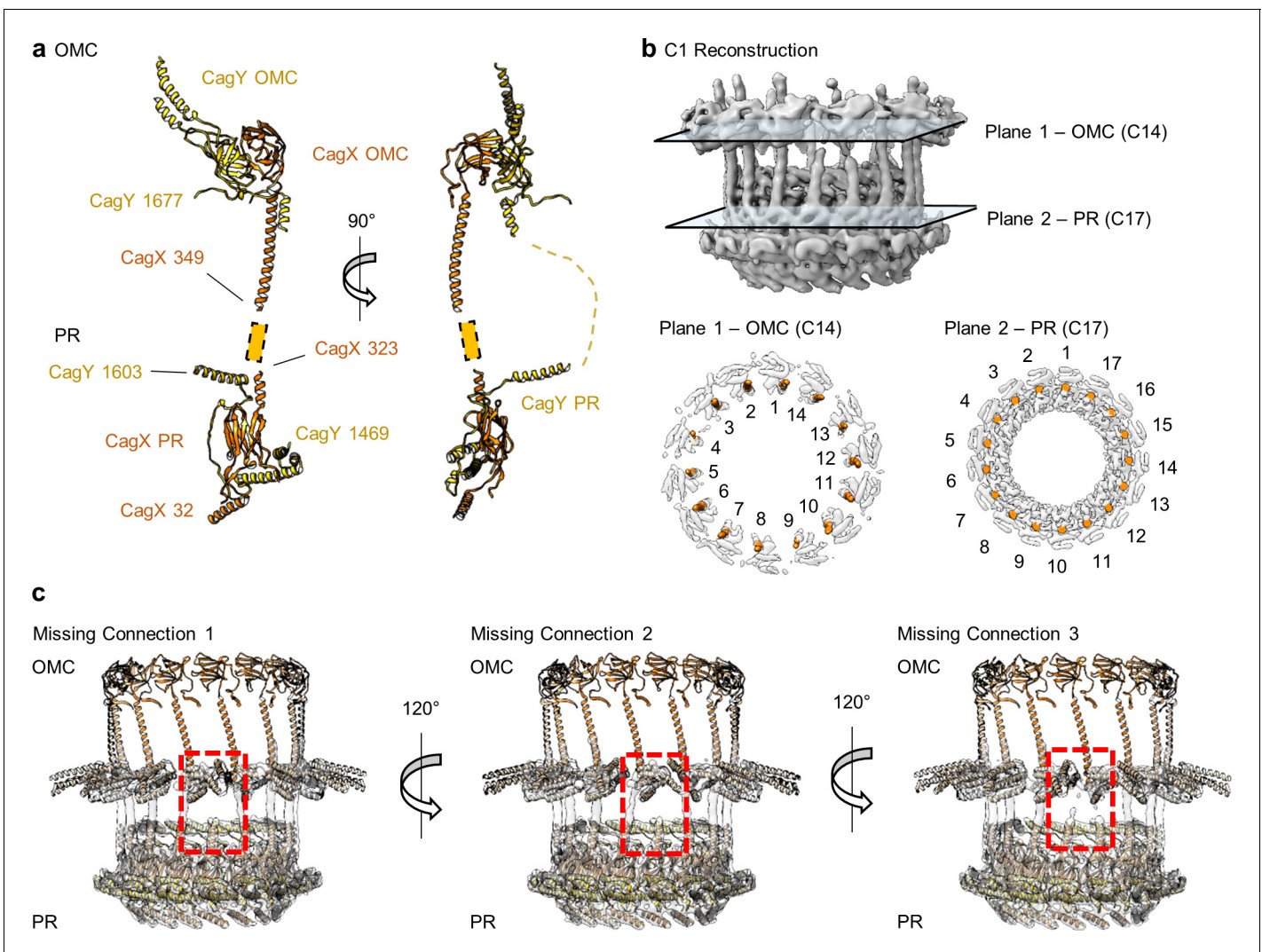

**Figure 9.** CagX and CagY span the symmetry mismatch between the OMC and PR. (a) Models of CagX and CagY within the OMC and PR asymmetric units. (b) In the asymmetric reconstruction from a focused refinement of the interface between the OMC and PR of the WT map, we note the presence of helical density predicted to be a portion of CagX (top). We have modeled 14 copies of CagX into density within the OMC (bottom, left) and 17 copies of CagX into the PR (bottom, right). (c) Though we observe clear connections between 14 subunits of the OMC and 14 subunits of the PR, three copies of CagX in the PR do not show any obvious connection to the OMC.

approach or by studies of recombinant proteins produced in *E. coli,* and are consistent with the structural observations reported here (*Busler et al., 2006*; *Kutter et al., 2008*; *Smart et al., 2017*). Importantly, the structural detail obtained from single particle cryo-EM analysis reveals the context of these interactions, allowing us to define which interactions occur within one asymmetric unit and which interactions occur between adjacent asymmetric units.

The structure that we present here provides an improved understanding of how the PR is assembled and highlights the presence of portions of CagX and CagY within the PR. We show that CagX and CagY assemble to form portions of both the OMC and PR, bridging the previously described symmetry mismatch within the core complex (*Chung et al., 2019*). Although the existence of CagX and CagY in both the OMC and PR seems inconsistent with the observed symmetry mismatch, it should be noted that this is not the first example of such a phenomenon. Indeed, a similar phenomenon has been detected in the Type Three Secretion System (T3SS) of *Salmonella,* in which one molecule (InvG) exists on both sides of a C15-C16 symmetry mismatch (*Hu et al., 2018*). Symmetry mismatch has been observed in multiple types of bacterial secretion systems or other large bacterial machines, including T2SSs, T3SSs, the *L. pneumophila* Dot/Icm T4SS, T6SSs, and the *Salmonella* flagellar motor (*Chernyatina and Low, 2019*; *Dix et al., 2018*; *Hu et al., 2018*; *Johnson et al., 2020*; *Park et al., 2020*), suggesting an important physiological function for this architecture. The mechanisms by which symmetry mismatch arises in these systems have not yet been determined, and the functional consequences of symmetry mismatch in these systems remain unclear. Interestingly, no symmetry mismatch has been observed in core complexes of minimized T4SS systems such as the *X. citri* T4SS, which contain 14 copies of both the N-terminal and C-terminal domains of VirB9 (*Sgro et al., 2018*).

When comparing the *H. pylori* Cag T4SS core complex to corresponding T4SS subassemblies in other species, there are striking differences. Most notably, the Cag T4SS core complex is much larger in size than core complexes in prototype systems, and it contains two species-specific components (Cag3 and CagM). The CagT, CagX and CagY components of the Cag T4SS are homologs of VirB7, VirB9 and VirB10 components of prototype systems, but there are marked differences in the sequences and structures. Finally, the Cag T4SS core complex includes a PR feature, which has not been detected in prototype systems. In the current study, we discovered that the portions of CagX and CagY within the PR are similar to the parts of VirB9 and VirB10 which reside in the I-layer of the *X. citri* T4SS (*Sgro et al., 2018*). Because of this relationship, it is perhaps appropriate to consider the PR of the Cag T4SS and the I-layer of the *X. citri* T4SS as analogous structures. However, there are several important distinctions. First, the difference in symmetry between the OMC and the PR of the *H. pylori* Cag T4SS contrasts the C14 symmetry observed throughout the *X. citri* T4SS core complex. Second, the position of the *X. citri* T4SS I-layer within the OMC (~115 Å below the outer membrane, as determined by the position of the membrane spanning helices) is similar to that of the I-layer of the Cag T4SS (positioned ~120 Å beneath the outer membrane, and composed of CagM). Therefore, in comparison to the Cag T4SS PR, the I-layer within the *X. citri* T4SS is positioned much closer to the OMC. The functional roles of these structures are currently unclear, but we speculate that the OMC I-layer and the PR represent structural expansions within the Cag T4SS that are associated with specific functions. Future investigations building on the current structural studies will be needed to develop a more complete understanding of the similarities and differences that exist among T4SSs.

## Materials and methods

### Key resources table

| Reagent type (species) or resource | Designation | Source or reference | Identifiers | Additional information |
|---|---|---|---|---|
| Strain, strain background (*Helicobacter pylori* 26695) | HA-CagF | PMID:26758182 | | produces HA epitope-tagged CagF |

*Continued on next page*

*Continued*

| Reagent type (species) or resource | Designation | Source or reference | Identifiers | Additional information |
|---|---|---|---|---|
| Strain, strain background (*Helicobacter pylori* 26695) | Δ*cag3*; HA-CagF | PMID:26758182 | | Δ*cag3* mutant, produces HA epitope-tagged CagF |
| Software, algorithm | Leginon | PMID:15890530 | | |
| Software, algorithm | MotionCor2 | PMID:28250466 | | |
| Software, algorithm | CTFFind4 | PMID:26278980 | | |
| Software, algorithm | cryoSPARC | PMID:28165473 | | |
| Software, algorithm | RELION | PMID:27685097 PMID:30412051 | | |
| Software, algorithm | Coot | PMID:20383002 | | |
| Software, algorithm | UCSF Chimera | PMID:15264254 PMID:29340616 | | |
| Software, algorithm | PHENIX | PMID:29872004 | | |
| Software, algorithm | DALI server | PMID:31263867 | | |

## Core complex purification

Cag T4SS complexes were purified from either a wild-type *H. pylori* strain or a Δ*cag3* mutant strain, each engineered to produce an epitope-tagged form of CagF, as described previously (*Chung et al., 2019*; *Frick-Cheng et al., 2016*). The resulting preparations were analyzed by SDS-PAGE and colloidal Coomassie blue staining, and the composition of the preparations was determined by LC-MS/MS analysis (*Frick-Cheng et al., 2016*; *Lin et al., 2020*).

## Cryo-EM data collection and map reconstruction

For cryo-EM, Cag T4SS core complex preparations purified from wild-type or Δ*cag3* mutant strains were applied to glow discharged Lacey 400 mesh copper grids (TED PELLA) coated with home-made ultra-thin (~2 nm) continuous carbon film or Quantifoil 2/2 200 mesh copper grids with ultra-thin (2 nm) continuous carbon film (Electron Microscopy Services), respectively. To increase particle density in suspended vitreous ice, 5 µL aliquots of the samples were applied three times, incubated for ~60 s after each application, and the grids were then washed with water to remove detergent (*Cheng et al., 2015*). The grids were vitrified by plunge-freezing in a slurry of liquid ethane using a Vitrobot Mark IV (Thermo Fisher Scientific) at 4°C and 100% humidity.

All the images were collected on a Thermo Fisher 300 kV Titan Krios electron microscope equipped with a Gatan K2 Summit Direct Electron Detector. The nominal pixel sizes for WT and ΔCag3 samples were 1.64 Å per pixel (x18,000 magnification) and 1.00 Å per pixel (x29,000 magnification), respectively. The total exposure time was 8 s, and frames were recorded every 0.2 s, resulting in a total accumulated dose of 59.0 e$^-$A$^{\circ-2}$ (for WT Cag T4SS) or 59.7 e$^-$A$^{\circ-2}$ (for ΔCag3 Cag T4SS). Different defocus ranges were used for each sample (−2.5 to −3.5 µm for WT and −1.5 to −3.5 µm for ΔCag3) as they had different ice-thicknesses. All the raw images for WT samples were identical to the datasets used in the previous study (*Chung et al., 2019*). In the previous study, approximately 20,000 raw micrographs were collected. Approximately 6,000 micrographs were selected and used for image processing steps in the current study, based on the quality of images.

All the image frames were first dose-weighted and aligned using MotionCor2 (*Zheng et al., 2017*). The contrast transfer function (CTF) values were then determined using CTFFind4 (*Rohou and Grigorieff, 2015*). Two different image processing software packages, cryoSPARC and

Relion3.0, were used for different types of processing (*Punjani et al., 2017*; *Zivanov et al., 2018*). For analysis of the WT Cag T4SS, 361,706 particles were picked from 5,980 micrographs using the template picker in cryoSPARC. The selected particles were then extracted with a 510 pixel box size (1.64 Å/pixel) and used to generate representative 2D classes. After the 2D classification, classes with clear secondary structural features and intact particles were retained (corresponding to 77,133 particles). The particles in these classes were then subjected to a 3D classification with a reference-free initial 3D model (*ab initio* model in cryoSPARC). The best 3D class (approximately 70,000 particles) was used as the reference for 3D refinement with C1 symmetry (lowpass filtered to 60 Å). Finally, a solvent mask and B-factor were applied to improve the overall features and resolution of the map, resulting in reconstruction of a 3D electron density map with a global resolution of 4.7 Å (*Figure 1—figure supplement 3*). For the ΔCag3 Cag T4SS dataset, 107,917 particles were selected using template picker in cryoSPARC and extracted with a 510 pixel box size (1.00 Å/pixel). After 2D and 3D particle filtering steps, approximately 12,000 particles were retained and subjected to 3D refinement with C1 symmetry. Subsequently, a solvent mask and B-factor were applied, resulting in reconstruction of a 3D map with 7.1 Å resolution (*Figure 1—figure supplement 2*).

The particle stacks used in 3D refinements of WT and ΔCag3 Cag T4SSs were then exported to Relion for further steps, such as per-particle CTF refinement and focused refinement. In the freezing step, the formation of relatively thick ice was necessary because of the very large size of the complexes (height,~470 Å). In addition, thin carbon support film (continuous ultra-thin carbon film) was used to overcome preferred particle orientation in vitrified ice. Within the conditions, different particles were situated at different heights within the field of view, causing per-particle defocus variations within a micrograph (*Zivanov et al., 2018*). Therefore, to reconstruct high resolution 3D models, per-particle defocus refinement and beam tilt estimation were applied, followed by 3D auto-refinements, which yielded a final resolution of 4.5 Å (for the WT Cag T4SS) and 5.3 Å (for the ΔCag3 Cag T4SS) with no symmetry applied.

For focused refinement of the WT OMC, signal subtraction for each individual particle containing the OMC was used with a soft mask. After signal subtraction and alignment-free focused 3D classification, one highly populated 3D class of OMC was produced (~47,000 particles). The best 3D class was then subjected to a masked 3D refinement, resulting in reconstruction of the 3D map at 3.6 Å resolution. Another round of per-particle defocus refinement and beam tilt estimation was applied, followed by another focused 3D refinement, which yielded a final resolution of 3.4 Å (*Figure 1—figure supplement 3*).

Similar steps were used for focused refinement of the ΔCag3 OMC and PR, starting with signal subtraction of the sub-complex for each particle. The subtracted particles were subjected to alignment-free focused 3D classification, and this produced one highly populated 3D class for each dataset (~7000 particles for the ΔCag3 OMC and ~10,000 particles for the ΔCag3 PR). The best 3D class was then subjected to a masked 3D refinement with C14 (ΔCag3 OMC) or C17 (ΔCag3 PR) symmetry, resulting in resolutions of 3.1 Å (OMC) and 3.0 Å (PR) (*Figure 1—figure supplement 2*). All resolutions described above were calculated using the gold-standard 0.143 FSC and visualized using UCSF Chimera and ChimeraX (*Goddard et al., 2018*; *Pettersen et al., 2004*).

## Model building and refinement

Models of the OMC were constructed within either the WT or the ΔCag3 Cag T4SS cryo-EM OMC maps using PDB-6OEE, PDB-6OEF, and PDB-6ODI as starting models (*Chung et al., 2019*). Each model was first docked into either the WT or the ΔCag3 Cag T4SS map using UCSF Chimera (*Pettersen et al., 2004*). The models were then iteratively built and refined within COOT (*Emsley et al., 2010*). After successive rounds of building, the models were subjected to real space refinement in Phenix with secondary structure and Ramachandran restraints applied (*Adams et al., 2010*; *Afonine et al., 2018*). The nonbonded weight applied during refinement was tuned to reduce steric clashing. Models of the PR were constructed de novo based on density reconstructed from the Δ*cag3* sample using COOT. These models were then iteratively refined in Phenix real space with secondary structure and Ramachandran restraints applied. Similarly, the nonbonded weight applied during refinement of the PR was tuned to reduce steric clashing. After refinement, this model was docked into a previously reported map corresponding to the PR of the WT Cag T4SS (EMD-20021) (*Chung et al., 2019*). This model was refined using a protocol similar to that which was described above. The quality of each model was determined by assessing how well the model fit the map using

model-map FSC calculations as well as per-residue correlation coefficients (CCs). All model building and refinement software was accessed through the SBGrid Consortium (*Morin et al., 2013*).

## Acknowledgements

We acknowledge the use of the University of Michigan cryo-EM facility, managed by M Su, A Bondy, and L Koepping, U-M LSI IT, and LSI and U-M BSI support. The work presented here was supported by NIH AI118932, CA116087, AI039657, GM103310, NIH 2T32DK007673, S10OD020011, and the Department of Veterans Affairs 5I01BX004447. A portion of the molecular graphics and analyses was performed with UCSF Chimera and ChimeraX developed by the Resource for Biocomputing, Visualization, and Informatics at UC-San Francisco, with support from NIH P41-GM103311.

## Additional information

### Funding

| Funder | Grant reference number | Author |
| --- | --- | --- |
| National Institute of Allergy and Infectious Diseases | AI118932 | Timothy L Cover<br>Melanie D Ohi<br>D Borden Lacy |
| National Institute of Allergy and Infectious Diseases | AI039657 | Timothy L Cover |
| National Cancer Institute | CA116087 | Timothy L Cover |
| National Institute of General Medical Sciences | GM103310 | Melanie D Ohi |
| National Institute of Diabetes and Digestive and Kidney Diseases | 2T32DK007673 | Michael J Sheedlo |
| National Institutes of Health | S10OD020011 | Melanie D Ohi |
| U.S. Department of Veterans Affairs | 5I01BX004447 | Timothy L Cover |

The funders had no role in study design, data collection and interpretation, or the decision to submit the work for publication.

### Author contributions

Michael J Sheedlo, Formal analysis, Validation, Investigation, Visualization, Writing - original draft, Writing - review and editing; Jeong Min Chung, Formal analysis, Validation, Investigation, Visualization, Writing - review and editing; Neha Sawhney, Resources, Investigation, Writing - review and editing; Clarissa L Durie, Investigation, Writing - review and editing; Timothy L Cover, Conceptualization, Resources, Supervision, Funding acquisition, Project administration, Writing - review and editing; Melanie D Ohi, Conceptualization, Formal analysis, Supervision, Funding acquisition, Validation, Investigation, Visualization, Project administration, Writing - review and editing; D Borden Lacy, Supervision, Validation, Investigation, Writing - review and editing

### Author ORCIDs

Michael J Sheedlo https://orcid.org/0000-0002-3185-1727
Jeong Min Chung https://orcid.org/0000-0002-4285-8764
Neha Sawhney http://orcid.org/0000-0002-4943-1018
Clarissa L Durie http://orcid.org/0000-0002-4027-4386
Timothy L Cover https://orcid.org/0000-0001-8503-002X
Melanie D Ohi http://orcid.org/0000-0003-1750-4793
D Borden Lacy https://orcid.org/0000-0003-2273-8121

### Decision letter and Author response
Decision letter https://doi.org/10.7554/eLife.59495.sa1

Author response https://doi.org/10.7554/eLife.59495.sa2

## Additional files

### Supplementary files
• Transparent reporting form

### Data availability

All cryo-EM data included in this manuscript are available through the Electron Microscopy Data Bank (EMD-20021, EMD-22081, EMD-22076, and EMD-22077). All models that were constructed from these data are available via the Protein Data Bank (PDB 6X6S, 6X6J, 6X6K, and 6X6L).

The following datasets were generated:

| Author(s) | Year | Dataset title | Dataset URL | Database and Identifier |
|---|---|---|---|---|
| Sheedlo MJ, Chung JM, Sawhney N, Durie CL, Cover TL, Ohi MD, Lacy DB | 2020 | Cryo-EM Structure of the Helicobacter pylori OMC | http://www.rcsb.org/structure/6X6S | RCSB Protein Data Bank, 6X6S |
| Sheedlo MJ, Chung JM, Sawhney N, Durie CL, Cover TL, Ohi MD, Lacy DB | 2020 | Cryo-EM Structure | http://www.rcsb.org/structure/6X6J | RCSB Protein Data Bank, 6X6J |
| Sheedlo MJ, Chung JM, Sawhney N, Durie CL, Cover TL, Ohi MD, Lacy DB | 2020 | Cryo-EM Structure | http://www.rcsb.org/structure/6X6K | RCSB Protein Data Bank, 6X6K |
| Sheedlo MJ, Chung JM, Sawhney N, Durie CL, Cover TL, Ohi MD, Lacy DB | 2020 | Cryo-EM Structure | http://www.rcsb.org/structure/6X6L | RCSB Protein Data Bank, 6X6L |
| Sheedlo MJ, Chung JM, Sawhney N, Durie CL, Cover TL, Ohi MD, Lacy DB | 2020 | HpT4SS OMC WT | https://www.ebi.ac.uk/pdbe/entry/emdb/EMD-22081 | Electron Microscopy Data Bank, EMD-22081 |
| Sheedlo MJ, Chung JM, Sawhney N, Durie CL, Cover TL, Ohi MD, Lacy DB | 2020 | HpT4SS PC WT | https://www.ebi.ac.uk/pdbe/entry/emdb/EMD-20021 | Electron Microscopy Data Bank, EMD-20021 |
| Sheedlo MJ, Chung JM, Sawhney N, Durie CL, Cover TL, Ohi MD, Lacy DB | 2020 | HpT4SS OMC Δcag3 | https://www.ebi.ac.uk/pdbe/entry/emdb/EMD-22076 | Electron Microscopy Data Bank, EMD-22076 |
| Sheedlo MJ, Chung JM, Sawhney N, Durie CL, Cover TL, Ohi MD, Lacy DB | 2020 | HpT4SS PR Δcag3 | https://www.ebi.ac.uk/pdbe/entry/emdb/EMD-22077 | Electron Microscopy Data Bank, EMD-22077 |

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

# Appendix 1

Appendix 1—table 1. Cryo-EM data collection and processing.

| | HpT4SS OMC WT | HpT4SS PC WT | HpT4SS OMC Δcag3 | HpT4SS PR Δcag3 |
|---|---|---|---|---|
| EMDB Accession Codes | EMD-22081 | EMD-20021 | EMD-22076 | EMD-22077 |
| **Data Collection and Processing** | | | | |
| Magnification | 18,000 | - | 29,000 | 29,000 |
| Voltage (kV) | 300 | - | 300 | 300 |
| Total Electron Dose (e⁻/Å²) | 59.0 | - | 59.7 | 59.7 |
| Defocus Range (μm) | −2.5 ~ −3.5 | - | −1.5 ~ −3.5 | −1.5 ~ −3.5 |
| Pixel Size (Å) | 1.64 | - | 1.0 | 1.0 |
| Processing Software | CryoSPARC/Relion | - | CryoSPARC/Relion | CryoSPARC/Relion |
| Symmetry | C14 | - | C14 | C17 |
| Initial Particles (number) | 361,706 | - | 107,917 | 107,917 |
| Final Particles (number) | 47,440 | - | 7337 | 10,477 |
| Map Sharpening B Factor | −64.34 | - | −47.40 | −52.23 |
| Map Resolution (Å) | 3.4 | - | 3.1 | 3.0 |
| FSC Threshold | 0.143 | - | 0.143 | 0.143 |
| **Model Refinement and Validation** | | | | |
| **Refinement** | | | | |
| Initial Model Used | 6OEF, 6OEE, 6ODI | 6X6L | 6OEF, 6OEE, 6ODI | 6OEF, 6OEE, 6ODI |
| **Model Resolution** | | | | |
| FSC (0.5) | 3.5 | 3.5 | 3.4 | 3.3 |
| FSC (0.143) | 3.3 | 3.4 | 3.1 | 3.0 |
| Model Composition (Residues) | 27,468 | | 7602 | 5083 |
| **Residues Modelled** | | | | |
| CagY | 1677–1820, 1851–1909 | 1469–1603 | 1677–1823, 1849–1910 | 1469–1603 |
| CagX | 349–514 | 32–130, 261–325 | 361–515 | 32–130, 261–325 |
| CagT-1 | 26–278 | - | 26–139, 165–175, 190–221, 286–307 | - |
| CagT-2 | 29–139, 178–238, 254–273 | - | - | - |
| Cag3-1 | 62–308 | - | - | - |
| Cag3-2 | 104–193 | - | - | - |
| Cag3-3 | 104–195 | - | - | - |
| Cag3-4 | 74–79, 94–181 | - | - | - |
| Cag3-5 | 78–181 | - | - | - |
| CagM-1 | 187–366 | - | | |
| CagM-2 | 187–365 | - | - | - |
| **Bond RMSD** | | | | |
| Bond Length (Å) | 0.011 | 0.006 | 0.011 | 0.007 |
| Bond Angle (°) | 1.490 | 0.616 | 1.279 | 0.775 |

*Continued on next page*

*Appendix 1—table 1 continued*

| | HpT4SS OMC WT | HpT4SS PC WT | HpT4SS OMC Δcag3 | HpT4SS PR Δcag3 |
|---|---|---|---|---|
| Validation | | | | |
| Molprobity Score | 2.15 | 2.51 | 1.82 | 2.59 |
| Clashscore | 9.64 | 13.41 | 8.21 | 11.66 |
| Ramachandran (%) | | | | |
| Favored | 94.0 | 95.9 | 94.5 | 96.4 |
| Allowed | 6.0 | 4.1 | 5.5 | 3.6 |
| Outliers | 0.0 | 0.0 | 0.0 | 0.0 |
| B Factor | 127.48 | 95.94 | 75.84 | 76.90 |
| PDB Accession Codes | 6X6S | 6X6J | 6X6K | 6X6L |

