## [Decision Letter]

**Acceptance summary:**

This is an important study which expands on the same group's previous contributions towards understanding the molecular organization of the core complex from the *H. pylori* Cag Type IV Secretion System (T4SS). In the present study the improved resolution of the spokes and outer ring enabled the authors to trace and assign much of the structure that was previously defined only at low resolution.

**Decision letter after peer review:**

Thank you for submitting your article "Cryo-EM reveals species-specific components within the *Helicobacter pylori* Cag type IV secretion system core complex" for consideration by *eLife*. Your article has been reviewed by three peer reviewers, and the evaluation has been overseen by a Reviewing Editor and John Kuriyan as the Senior Editor. The reviewers have opted to remain anonymous.

The reviewers have discussed the reviews with one another and the Reviewing Editor has drafted this decision to help you prepare a revised submission.

Summary:

This is an important study that will be received with great interest. It expands on the group's previous contributions towards understanding the molecular organization of the core complex from the *H. pylori* Cag T4SS. The group presents single particle cryoEM analyses of a core complex from a Cag3-deficient strain (*Δcag3*) and re-analyzed data previously collected for wild-type core complexes to obtain higher resolution maps. Their previous publication in *eLife* (Chung et al., 2019) described the general features of the structure: an outer membrane complex (OMC or OMCC) with 14-fold symmetry and a periplasmic ring complex (PR or PRC) with 17-fold symmetry. The previous OMC structure presented an inner and outer ring connected by "spokes", with the inner ring consisting of 14 copies each of CagT and the C-terminal domains of CagX and CagY. In the present study the improved resolution of the spokes and outer ring enabled the authors to trace the partial chains of a 14 copies of {a second CagT subunit, 5 Cag3 subunits and 2 CagM subunits}. This results in a 1:1:2:2:5 CagY:CagX:CagT:CagM:Cag3 molar ratio for the OMC. Also, in the previous study the PRC was modeled as 17 copies of a polyalanine chain of unknown sequence, speculated at the time to be made of CagM. In the present study this the PRC is in fact shown to consist of 17 copies of the N-terminal domain of CagX plus 17 copies of a segment from CagY. Finally, the authors present experimental evidence that suggests a possible explanation for the symmetry mismatch between the OMC and PR that both contain portions of CagX and CagY proteins.

Essential revisions:

1) Atomic models of the 4 structures were provided but the corresponding EM maps were not. Therefore, this reviewer was not able to determine how well the sequence matches the density in the previously ambiguous regions of the map, especially for the CagM and Cag3 subunits in the OMCC and the CagY fragments in the PRC. Supplementary figures showing the quality of the EM map in these regions should be included.

2) The suggestion that the symmetry mismatch may involve the proteolysis of CagX and/or CagY proteins (so that 17 copies contribute to the PR while only 14 contribute to the OMC) needs some experimental support. One suggestion is to use SDS-PAGE of the samples: 3/17 = 17% proteolysis should be detectable. In any case, such electrophoretic analysis of the samples should be provided as supplementary material.

3) The reviewers felt that additional evidence was also required for the stoichiometry and organisation of Cag3, CagM and CagX/Y. Suggestions for how to do this involved cross-linking mass spec, but the reviewers were open to other approaches. The comments from the reviewers were as follows:

– The number of Cag3 and their organization should be probed by cross-linking mass spectrometry.

– More importantly, same applies to CagM and CagX/CagY which belong to the two subcomplexes.

– The proposed 1:1:2:2:5 stoichiometry for X:Y:M:T:3 should also be confirmed by alternate methods.

---

## [Author Response]

Essential revisions:1) Atomic models of the 4 structures were provided but the corresponding EM maps were not. Therefore, this reviewer was not able to determine how well the sequence matches the density in the previously ambiguous regions of the map, especially for the CagM and Cag3 subunits in the OMCC and the CagY fragments in the PRC. Supplementary figures showing the quality of the EM map in these regions should be included.

We agree and regret that the maps could not be compressed sufficiently for upload. We are happy to provide the maps, but we also want the publication to demonstrate the quality of both the models and the maps in all regions for which we have determined structures. We have added six supplementary figures to Figure 2 (Figure 2—figure supplements 1-6). These new figures show the quality of the maps and models within the OMC/PR for both the WT and ΔCag3 complexes. We have also mentioned this in the Materials and methods section (subsection “Model Building and Refinement”). Additionally, we have added a supplementary figure (Figure 4—figure supplement 1) discussing CagT-1 and CagT-2 interactions, which includes a panel (panel F) highlighting the quality of density that is observed at these interfaces. We have also added a panel to another new supplementary figure (Figure 4—figure supplement 2) to show the quality of the map and models of all the C-terminal helices of CagT-1 and CagT-2 in both the WT and ΔCag3 maps. To address concerns regarding the register and the quality of density in the peripheral copies of Cag3, we have included a supplementary figure (Figure 6—figure supplement 1) that depicts the density in these areas, including examples that indicate the presence of a strand-exchange with adjacent molecules (panel F). Finally, we have revised panels in Figure 7 (panels A and B) to better demonstrate the quality of data in the portions of the map corresponding to CagM-1 and CagM-2. Collectively, these additional figures provide readers with a better understanding of the quality of the EM maps and provide reassurance that the maps closely match the assigned sequences.

2) The suggestion that the symmetry mismatch may involve the proteolysis of CagX and/or CagY proteins (so that 17 copies contribute to the PR while only 14 contribute to the OMC) needs some experimental support. One suggestion is to use SDS-PAGE of the samples: 3/17 = 17% proteolysis should be detectable. In any case, such electrophoretic analysis of the samples should be provided as supplementary material.

We suggested proteolysis as one of several possible models that could account for 14 copies of CagX/CagY in the OMC and 17 copies in the PR. In line with suggestions from the reviewers, we have revised the manuscript to clarify that there are multiple possible explanations for how symmetry mismatch can arise.

“It is not clear if the density corresponding to these copies of CagX and CagY cannot be traced due to inherent flexibility within the respective C-terminal domains, if these additional protomers represent truncated versions of CagX and CagY, or if they represent uncharacterized structural homologs.”

We also comment that despite the presence of symmetry mismatch in several types of secretion systems, the mechanisms by which symmetry mismatch arises have not been established.

“Although the existence of CagX and CagY in both the OMC and PR seems inconsistent with the observed symmetry mismatch, it should be noted that this is not the first example of such a phenomenon. […] The mechanisms by which symmetry mismatch arises in these systems have not yet been determined, and the functional consequences of symmetry mismatch in these systems remains unclear.”

It would be almost impossible for us to confidently detect or exclude ~17% proteolysis of CagX and/or CagY using SDS-PAGE analysis. In response to the last sentence of this comment, we have included SDS-PAGE and mass spectrometric analyses of the wild-type and ∆*cag3* core complex samples in Figure 1—figure supplement 4.

3) The reviewers felt that additional evidence was also required for the stoichiometry and organisation of Cag3, CagM and CagX/Y. Suggestions for how to do this involved cross-linking mass spec, but the reviewers were open to other approaches. The comments from the reviewers were as follows:– The number of Cag3 and their organization should be probed by cross-linking mass spectrometry.– More importantly, same applies to CagM and CagX/CagY which belong to the two subcomplexes.– The proposed 1:1:2:2:5 stoichiometry for X:Y:M:T:3 should also be confirmed by alternate methods.

Presumably these comments arise in part because of reviewers’ uncertainty about the quality of the EM maps. In the revised manuscript, we show additional data that further describe the basis for our high level of confidence in matching protein sequences to specific regions of the EM maps and determining the register of individual Cag proteins (see response to point 1 above). We have previously attempted cross-linking mass spectrometry analysis of the Cag T4SS core complex samples without success. The difficulty is probably attributable to multiple factors, including the low concentration of T4SS complexes extracted from *H. pylori*, non-specific aggregation of the complexes, and the complexity of the structures (each of which contains 5 different proteins in multiple copies, with a total of 154 proteins per complex, as proposed in our model). Even under the best of circumstances, cross-linking mass spectrometry would not corroborate the number of Cag3 proteins or provide information about the overall stoichiometry. On the other hand, numerous experiments have been done previously by multiple groups (including one of the author’s labs) to define protein-protein interactions among Cag core complex components. The cryo-EM results reported in the current manuscript match very well with the results of these previous studies (including yeast two-hybrid experiments and studies of recombinant proteins produced in *E. coli*). In our revised manuscript, we now discuss the previous studies and explain how they support the cryo-EM findings pertaining to the organization of Cag3, CagM, CagT, and CagX/Y.

“The current analysis also provides a better understanding of how the components of the Cag T4SS are arranged, highlighting important interactions among the newly described components. […] Importantly, the structural detail obtained from single particle cryo-EM analysis reveals the context of these interactions, allowing us to define which interactions occur within one asymmetric unit and which interactions occur between adjacent asymmetric units.”.